# Discovery of Triple Inhibitors of Both SARS-CoV-2 Proteases and Human Cathepsin L

**DOI:** 10.3390/ph15060744

**Published:** 2022-06-13

**Authors:** Ittipat Meewan, Jacob Kattoula, Julius Y. Kattoula, Danielle Skinner, Pavla Fajtová, Miriam A. Giardini, Brendon Woodworth, James H. McKerrow, Jair Lage de Siqueira-Neto, Anthony J. O’Donoghue, Ruben Abagyan

**Affiliations:** 1Skaggs School of Pharmacy and Pharmaceutical Sciences, University of California San Diego, La Jolla, CA 92093, USA; imeewan@ucsd.edu (I.M.); deskinner@health.ucsd.edu (D.S.); pfajtova@health.ucsd.edu (P.F.); mgiardini@health.ucsd.edu (M.A.G.); btwoodworth@health.ucsd.edu (B.W.); jmckerrow@health.ucsd.edu (J.H.M.); jlagedesiqueiraneto@health.ucsd.edu (J.L.d.S.-N.); ajodonoghue@health.ucsd.edu (A.J.O.); 2Department of Chemistry and Biochemistry, University of California San Diego, La Jolla, CA 92093, USA; 3School of Biological Sciences, University of California San Diego, La Jolla, CA 92093, USA; jakattoula@health.ucsd.edu (J.K.); jykattou@ucsd.edu (J.Y.K.)

**Keywords:** COVID-19 drug candidates, multiple protease inhibitors, disulfiram, thiuram disulfide, dithiobis-(thioformate), SARS-CoV-2 main protease, SARS-CoV-2 papain-like protease, cathepsin L, transmembrane serine protease 2 TMPRSS2, COVID-19

## Abstract

One inhibitor of the main SARS-CoV-2 protease has been approved recently by the FDA, yet it targets only SARS-CoV-2 main protease (Mpro). Here, we discovered inhibitors containing thiuram disulfide or dithiobis-(thioformate) tested against *three* key proteases involved in SARS-CoV-2 replication, including Mpro, SARS-CoV-2 papain-like protease (PLpro), and human cathepsin L. The use of thiuram disulfide and dithiobis-(thioformate) covalent inhibitor warheads was inspired by an idea to find a better alternative than disulfiram, an approved treatment for chronic alcoholism that is currently in phase 2 clinical trials against SARS-CoV-2. Our goal was to find more potent inhibitors that target both viral proteases and one essential human protease to reduce the dosage, improve the efficacy, and minimize the adverse effects associated with these agents. We found that compounds coded as RI175, RI173, and RI172 were the most potent inhibitors in an enzymatic assay against SARS-CoV-2 Mpro, SARS-CoV-2 PLpro, and human cathepsin L, with IC_50_s of 300, 200, and 200 nM, which is about 5-, 19-, and 11-fold more potent than disulfiram, respectively. In addition, RI173 was tested against SARS-CoV-2 in a cell-based and toxicity assay and was shown to have a greater antiviral effect than disulfiram. The identified compounds demonstrated the promising potential of thiuram disulfide or dithiobis-(thioformate) as a reactive functional group in small molecules that could be further developed for treatment of the COVID-19 virus or related variants.

## 1. Introduction

The severe acute respiratory syndrome coronavirus (SARS-CoV) and the Middle East respiratory syndrome coronavirus (MERS-CoV) are members of Coronaviridae that can cause fatal respiratory diseases and are rapidly transmitted, but their outbreaks were far from pandemic in scale. In December 2019, a new coronavirus known as severe acute respiratory syndrome coronavirus 2 (SARS-CoV-2) was identified in Wuhan, China, that causes coronavirus disease 2019 (COVID-19) [1,2]. COVID-19 has swept the world since early 2020; at the time of writing this paper, over 6.23 million deaths and 513 million infections have occurred worldwide, causing a global pandemic [3]. Heavy priority has been placed on finding an effective antiviral drug, and several viral targets have been considered.

Two important viral targets controlling the production of functional proteins by SARS-CoV-2, a positive sense RNA virus, include main protease (Mpro) and papain-like protease (PLpro) encoded by the viral genome [4]. A frameshift between open reading frame 1a (ORF1a) and open reading frame 1b (ORF1b) enables the production of two polypeptides: Polypeptide 1a (pp1a) and polypeptide 1b (pp1ab). The viral proteases Mpro and PLpro process pp1a and pp1ab into 16 non-structural proteins necessary for viral assembly and replication as depicted in Figure 1. While inhibitors of individual proteases have been suggested [5,6], there are obvious benefits from the inhibition of *both* viral proteases to limit overall SARS-CoV-2 viral replication.

Moreover, the SARS-CoV-2 entry mechanism relies on two *human* proteases: the transmembrane serine protease 2 (TMPRSS2) and endosomal cathepsin [7,8,9]. TMPRSS2 resides on cell membranes and facilitates the fusion process. In addition, SARS-CoV-2 takes advantage of endosomal cathepsin L [10,11]. Both viral and human proteases are attractive targets for the design of anti-SARS-CoV-2 drugs. Our goal was to target *multiple* proteases that are necessary for the viral replication process with a *single* small molecule inhibitor. This approach is challenging, but it may result in better antiviral drugs that are less sensitive to viral escape mutations resulting in drug resistance.

Currently, two antiviral drugs have received the FDA EUA for the treatment of COVID-19 infection: Molnupiravir and Paxlovid/Nirmatrelvir, with the latter targeting SARS-CoV-2 Mpro. In addition, several peptidomimetic compounds exhibiting effective inhibition against individual SARS-CoV-2 proteases have been identified [12,13,14] and many drugs have been investigated for repurposing toward the COVID-19 treatment [15]. However, our interests lie in covalent inhibitors that inhibit multiple useful targets related to COVID-19. One interesting compound was disulfiram, which contains a thiuram disulfide covalent warhead at the center of the molecule. Disulfiram is a currently approved drug commonly prescribed as an aversive treatment of chronic alcoholism. Disulfiram primarily acts as an irreversible inhibitor of aldehyde dehydrogenase by binding to a cysteine residue within the active site. The binding disrupts the breakdown of alcohol, causing the development of resistance to alcohol consumption [16].

Beyond the treatment of alcoholism, the distinct structure of disulfiram, containing the thiuram disulfide covalent warhead, is viewed as a key component of the drug’s activity in SARS-CoV-2 and is maintained as a point of interest in drugs targeting viral proteases. Previously, disulfiram was shown to inhibit both MERS-CoV and SARS-CoV PLpro with 14.6 and 24.1 μM IC_50_ values, respectively [17,18,19]. It was shown that disulfiram may act at the active site of the SARS-CoV PLpro, creating a covalent adduct with Cys111 that implied possible future uses of the drug in treatment against coronaviruses [20]. More recently, disulfiram was characterized to display antiviral activity against multiple viral proteases, including SARS-CoV-2 Mpro (IC_50_: 9.35 µM) and SARS-CoV-2 PLpro (IC_50_: 6.9 µM) and several other viral cysteine proteases [18].

Furthermore, disulfiram has been characterized to block the pore formation of gasdermin D (GSDMD) through covalent targeting of cysteine 191, counteracting IL-β release and inflammation [21]. In phosphoglycerate dehydrogenase (PHGDH) and caspase 1, disulfiram covalently binds to cysteine residues, which ultimately blocks the release of cytokines [22,23]. Further characterized targets that have been shown to be inhibited by disulfiram are displayed in Table 1. With knowledge of this counteraction to inflammation and other favorable outcomes of disulfiram, it is currently in phase 2 clinical trials, being identified as a potential therapeutic target for SARS-CoV-2 [24,25].

Previously, some derivatives of disulfiram, designated here with an RIxxx label, have been found to inhibit certain cysteine-containing enzymes, including proteases. Due to the efficacy of disulfiram in clinical trials and the previously identified activities of these analogs, all commercially available drugs in this class were tested on the SARS-CoV-2 papain-like and main protease to determine more effective inhibitors that take advantage of disulfiram’s mechanism of binding to an active-site cysteine. We studied thiuram disulfide or dithiobis-(thioformate)-containing compounds and varied the functional groups. It was discovered that most analogs had either a similar or greater inhibition for both SARS-CoV-2 Mpro and PLpro in comparison to that of disulfiram. Moreover, three of them inhibited human cathepsin L essential for the SARS-CoV-2 infectious cycle. These triple target inhibitors of both SARS-CoV-2 proteases and human cathepsin L may lead to a new multi-target strategy for COVID-19 treatment.

## 2. Results

### 2.1. SARS-CoV-2 Mpro and PLpro Activity Assay

Disulfiram, a known irreversible covalent inhibitor targeting cysteine proteases [34], was tested along with related but different compounds that contain thiuram disulfide or dithiobis-(thioformate), a necessary functional group for covalent binding to cysteine protease targets [35]. We selected six thiuram disulfide analogs with different N-substituents and one dithiobis-(thioformate), RI171 to RI177, to test against SARS-CoV-2 Mpro and PLpro. We performed fluorometric enzymatic assays to show the activities of Mpro and PLpro in the presence of the selected compounds. The assays were performed using a FRET-based peptide substrate: Z-RLRGG-AMC at 50 µM for Mpro and PLpro, respectively. As shown in Figure 2 and Table 1, the inhibitory activities of RI172 and RI173 against Mpro with the half-maximum inhibitory concentration (IC_50_) of 0.6 ± 0.1 and 0.6 ± 0.1 µM, respectively, showed a 2.5-fold improvement from disulfiram, with an IC_50_ value of 1.5 ± 0.1 µM (2.1 ± 0.3 µM reported value [18]), while RI174, RI171, and RI177 showed slightly weaker inhibition effects compared to disulfiram. In addition, four of the thiuram disulfide compounds: RI173, RI172, RI174, and RI171, showed significant potency improvement against PLpro, with IC_50_ of 0.2 ± 0.1, 0.7 ± 0.0, 0.6 ± 0.0, and 1.9 ± 0.1 µM, respectively, compared to the reported IC_50_ of disulfiram of 3.8 µM (6.9 ± 4.2 µM reported value [18], especially for RI173, which showed a 19-fold improvement over disulfiram). These results suggest that thiuram disulfide is a necessary moiety to interact covalently with cysteine active residues of Mpro and PLpro.

The replacement of diethylamine groups in disulfiram with morpholine, pyrrolidine, dimethylamine, and 4-methylpiperazine in RI173, RI172, RI174, and RI171, respectively, moderately enhanced the potency of inhibitors against Mpro and PLpro while substitution by diisopropylamine in RI177 showed the preservation of the inhibition of Mpro but complete loss of the inhibitory activity (IC_50_ > 50 µM) against PLpro. Furthermore, the presence of the N-methylaniline moiety in RI177 entirely lost the inhibition potency (IC_50_ > 50 µM) in both Mpro and PLpro. These results suggest that the substitution of bulky and flat moieties at terminal N-substituents is unfavorable for the improvement of the inhibition against SARS-CoV-2 proteases for thiuram disulfide analogues. It is especially interesting that RI175, which contains dithiobis-(thioformate), unlike the rest of the compounds in this series, showed potent inhibitory activity against both Mpro and PLpro: 0.3 ± 0.1 and 0.6 ± 0.1 µM, respectively. Thiuram disulfide and dithiobis-(thioformate) are known to react strongly to cysteine proteases [36,37]. Therefore, the small molecules containing this warhead are chemically susceptible to nonspecific competing cysteine compounds, such as dithiothreitol (DTT) in solution buffer. The concentration of DTT, a redox agent, in previous studies in our comparison assay may well exceed the toxic threshold. Hence, DTT, in contrast to disulfiram, is not suitable for systemic administration and is useful as a potential therapeutic agent. DTT, in essence, destroys the assay because it degrades thiuram disulfide or dithiobis-(thioformate) prior to their disulfide interactions with the cysteine proteases (Mpro and PLpro) of SARS-CoV-2 and human cathepsin L or changes the baseline protease activity by binding to active site cysteines. Therefore, as expected, the inhibition of thiuram disulfide or dithiobis-(thioformate) analogues against SARS-CoV-2 Mpro and PLpro dropped significantly in the presence of 1 mM of DTT as shown in Appendix A. In fact, the DTT interference with the inhibitors confirmed the mechanism of action of these analogues as they reacted covalently to multiple cysteine proteases. The efficacy of thiuram disulfide or dithiobis-(thioformate) analogues in vivo should not be affected by competing DTT or glutathione, as illustrated by the clinical efficacy of disulfiram [24].

### 2.2. Inhibition of Human Cathepsin L Protease with Identified Inhibitors of Viral Proteases

Beside Mpro and PLpro, there are host proteases, including TMPRSS2, cathepsin L, and furin, that are necessary for SARS-CoV-2 viral replication. We selected both the endosomal cathepsin L and TMPRSS2 proteases that facilitate viral entry into the cell. SARS-CoV-2 binds to ACE2 on the cell surface and can enter cells either by single or dual activation mechanisms from cathepsin L in the endosome and TMPRSS2 on the cell surface [10]. In the previous section, several selected thiuram disulfide or dithiobis-(thioformate) analogues have shown excellent broadened inhibitory activity against both SARS-CoV-2 viral proteases. The mechanism of action of thiuram disulfide or dithiobis-(thioformate) analogues is to bind to sulfhydryl groups of cysteine active residues of Mpro and PLpro, becoming their oxidized cysteine form (Figure 3). Since cathepsin L contains cysteine as an active residue, similar to Mpro and PLpro [36,37], we tested whether these compounds could also target cathepsin L, perhaps providing small molecules that target multiple key proteins for the viral replication process, producing a more favorable outcome for SARS-CoV-2 treatment. Thus, we also evaluated the inhibition activity of the selected compounds against 1 nM of cathepsin L using 30 µM of Z-Phe-Arg-AMC as a fluorogenic substrate; the dose–response curves of each compound are shown in Figure 4. The IC_50_ values of compounds RI173 and RI175 (1.7 ± 0.6 and 1.9 ± 0.6 µM) showed a slight improvement from disulfiram (2.2 ± 1.4 µM). RI172 showed excellent potency (0.2 ± 0.0 µM) and significant enhancement: approximately 11.5 times more potent than disulfiram. Compounds RI171 and RI174 exhibited less potency compared to disulfiram, but they are still in the acceptable range. On the other hand, thiuram disulfide analogs connected to the bulky substituents isopropyl and methyl-di-phenyl, RI176 and RI177, respectively, did not seem to have any inhibitory effect on cathepsin L. This suggests that small alkyl or heterocyclic substituents on thiuram disulfide or dithiobis-(thioformate) analogs are more favorable than bulky substituents for strong inhibition of cathepsin L.

However, inactivation of cathepsin L does not completely prevent viral entry into the cell as SARS-CoV-2 utilizes TMPRSS2 as an alternative entry pathway [10]. We tested our compounds against 30 nM of human TMPRSS2 using fluorogenic substrate 100 µM Boc-Gln-Ala-Arg-AMC; however, there was no inhibitory activity of any compound up to 100 μM against serine protease TMPRSS2 (Appendix A). This suggests cysteine proteases’ specificity for the thiuram disulfide or dithiobis-(thioformate) analogues. According to these results, we found triple inhibitors that target SARS-CoV-2 Mpro, SARS-CoV-2 PLpro, and human cathepsin L.

### 2.3. Indirect Effect in COVID-19 Treatment through Inhibition of Thrombin

Previously, it was shown that several thiuram disulfide or dithiobis-(thioformate) compounds have high potency against Mpro and PLpro of SARS-CoV-2 and human cathepsin L. Yet, we desired to further investigate whether these compounds also inhibited other proteases related to the SARS-CoV-2 infection to provide further benefits in COVID-19 treatment and, most importantly, spare essential human proteases involved in blood clotting. Severe pneumonia, acute inflammation, and blood clots are common complications found in some severe COVID-19 infections [38,39,40,41,42]. We were happy to find that thrombin (1 nM) was not affected, even at the 100 μM concentration of our analogs. Appendix A illustrates the absence of significant inhibition of thrombin.

### 2.4. SARS-CoV-2 Infectivity Cells-Based Assay

Thiuram disulfide and dithiobis-(thioformate) inhibit three proteases that function in various critical stages of viral replication, including viral entry and viral maturation. Targeting multiple proteases may have a therapeutic advantage over drugs that inhibit a single protease as it may reduce the emergence of drug-resistant mutants. Therefore, we tested these triple-target compounds in a SARS-CoV-2 Huh 7.5.1 infection model. Cells were infected with SARS-CoV-2 in an MOI of 1.0 and treated with thiuram disulfide or dithiobis-(thioformate) for 48 h. The antiviral efficacy of the compounds was tested as a half dilution in the range of 60 to 0.08 µM. We employed an immunofluorescence technique to visualize the number of viruses and host cells in the presence of these inhibitors. The half maximal effective concentration (EC_50_) and cytotoxic concentration (CC_50_) were determined (Figure 5 and Table 1). The relative antiviral activity was normalized to the untreated control (0% inhibition) and non-infected control (100% inhibition). A parameter related to the therapeutic index (TI) of each compound was calculated as the ratio of CC_50_ over EC_50_. The results indicated that RI173 exhibits antiviral activity in a narrow range from approximately 100 to 200 nM. At a slightly higher concentration, RI173 quickly became too toxic as % cell viability dropped rapidly. The therapeutic ratio of RI173 was 3, which is an *improvement compared to disulfiram*. However, compounds RI172, RI174, and RI177 were too toxic for the host cells in the selected range. The rest of the compounds failed to show antiviral efficacy in the expected range. In addition, the available data for the toxicity profiles of disulfiram, RI173, RI175, RI174, and RI171 as the median lethal dose (LD_50_) for the oral doses in mice is shown (Table 1) [27,28,29,30,31,37]. The LD_50_ of RI173 is approximately 1.6 times higher than the LD_50_ of disulfiram, suggesting less toxicity of this drug in animal models. RI175 and RI174 have slightly lower LD_50_ values compared to disulfiram; however, there is no available LD_50_ for RI172, RI175, and RI176. Although there is only one compound for which our results show a small improvement from the approved drug, disulfiram, these results provide the proof of concept of targeting multiple functional proteins involving SARS-CoV-2 viral replication using thiuram disulfide or dithiobis-(thioformate) compounds for the improvement of COVID-19 treatment.

## 3. Discussion

The current treatment of COVID-19 has relied on available approved medicines such as an Mpro inhibitor nirmatrelvir, α-interferon, repurposed anti-HIV medication lopinavir, ritonavir, and remdesivir [43,44,45,46]. More specific and effective therapies are necessary for post infection treatment. Most antiviral research is focused on specific single target inhibitors and are subject to the emergence of drug resistance. On the other hand, targeting several proteins essential for the viral cycle can reduce this vulnerability. Disulfiram has been identified as a potent SARS-CoV-2 Mpro and PLpro inhibitor and is currently in phase two of a clinical trial study for COVID-19 treatment. Moreover, disulfiram also covalently targets multiple enzymes, including aldehyde dehydrogenase (ALDH2) [16], dopamine beta hydroxylase (DBH) [26], gasdermin D (GSDMD) [21], pyruvate dehydrogenase kinase 1 (PDK1) [27], ubiquitin E3 ligase breast cancer-associated protein 2 (BCA2) [28], human monoacylglycerol lipase (hMGL) [30], and human cathepsin L. A multi-target pharmacological approach against enzymes that are directly or indirectly associated with SARS-CoV-2 infection may improve the treatment of severe viral infection and reduce the emergence of drug-resistant variants. Here, we showed that the thiuram disulfide or dithiobis-(thioformate) analogs can target key proteases in SARS-CoV-2 replication, namely the viral Mpro and PLpro enzymes in addition to human cathepsin L in enzymatic assays. We found that these compounds improve the inhibitory effectiveness against Mpro, PLpro, and cathepsin L, when compared to disulfiram, respectively. The cells-based infectivity assays indicated that RI173, the best compound in this series, showed a range of antiviral activity without killing the host cells while disulfiram and the rest of the compounds in this class tend to be high in toxicity at the concentrations at which antiviral efficacy was observed. This may indicate possible off-target effects of disulfiram and its various analogs. The required concentration range of RI173 is lower; hence, it is likely to be beneficial for treatment against COVID-19. Although the suggested compounds do not inhibit another essential host serine protease, TMPRSS2, a combination therapy with TMPRSS2 inhibitors, such as nafamostat, camostat, and gabexate mesylate, could improve the treatment efficacy in patients who have COVID-19. These compounds have other activities that may be beneficial. In addition to targeting SARS-CoV-2 Mpro and PLpro, human cathepsin L, and aldehyde dehydrogenase, disulfiram and RI173, the best compound in this series, is a known inhibitor of pyruvate dehydrogenase kinase 1 (PDK1) [27], which functions as a switch from mitochondrial respiration to aerobic glycolysis. This process is called the “Warburg effect” and is thought to enhance malignancy [37,39]. SARS-CoV-2, similar to MERS-CoV, has a replication process that is promoted by the Warburg effect via an increase in the production of the required resources for viral replication during the aerobic glycolysis enhancement [47,48]. By blocking the activity of PDK1, RI173 and other thiuram disulfide or dithiobis-(thioformate) compounds in this series will help diminish SARS-CoV-2 viral replication, which is favorable for COVID-19 treatment. In addition, for most COVID-19 patients, high production of proinflammatory cytokines is observed. These cytokines, in excessive amounts, can lead to acute lung injury and death [39,40]. Luckily, thiuram disulfide in disulfiram has been shown to covalently target Cys191 of gasdermin D (GSDMD) and Cys116 of phosphoglycerate dehydrogenase (PHDGH), which reduces the release of inflammatory cytokines [21,49]. The same effect may also be assumed that dithiobis-thioformated analogues.

The results demonstrated that thiuram disulfide- or dithiobis-(thioformate)-containing compounds can directly inhibit Mpro and PLpro viral proteases and human cathepsin L, and provide indirect benefits via disruption of viral replication processes and the reduction of excess inflammatory cytokines through PDK1, GSMD, and PHDGH inhibition. The main concerns of the thiuram disulfide or dithiobis-(thioformate) moiety include the high level of toxicity and non-specificity towards cysteine proteases, which may cause side effects [18]. However, the approval of disulfiram illustrates that these obstacles are not unsurmountable. Further improvement of the identified analogues may be needed.

## 4. Materials and Methods

### 4.1. Computational Modeling

High-resolution X-ray crystal structures of SARS-CoV-2 PLpro, Mpro, and human cathepsin L from Protein Data Bank (PDB code: 5XBG and 6WX4, respectively) were converted into 3D models for covalent docking of the studied compounds. The RI173 protein-docking in both PLpro and Mpro and a conveyed mechanism of action of dithiobis-(thioformate) analogs are shown in Figure 3. The ligand docking pose and its binding affinity were estimated by docking scores, representing the binding Gibbs free energy. The algorithm for the sampling 3D conformation of ligands and pockets was generated randomly based on biased probability Monte Carlo. All covalent docking simulations, scoring functions, and pharmacokinetic properties were also predicted from ICM-Pro v3.9-2b Molsoft LLC. (San Diego, CA, USA) [50,51].

### 4.2. Compounds and Reagents

Disulfiram was purchased from Fisher Scientific (Hampton, NH, USA). RI171 and RI172 were purchased from MolPort (Rīga, Latvia) while RI173 was purchased from Ambeed (Arlington Heights, IL, USA). RI174, RI175, RI176, and RI177 were obtained from Sigma-Aldrich (St. Louis, MO, USA). All compounds were dissolved in dimethyl sulfoxide. All solvents were reagent grade, and all reagents were purchased from Sigma-Aldrich (St. Louis, MO, USA).

### 4.3. Recombinant Protein and Substrates

The recombinant SARS-CoV-2 main protease (Mpro) was expressed using the Mpro plasmid provided by Rolf Hilgenfeld [52] and purified as previously described [12,52]. Recombinant proteases were purchased from following vendors: SARS-CoV-2 PLpro (Acro Biosystems, Newark, DE, USA), Thrombin (R & D Systems, Minneapolis, MN, USA), TMPRSS2 (Cusabio Technology LLC., Houston, TX, USA), and human cathepsin L (R & D Systems, Minneapolis, MN, USA). Protease substrates were purchased from the following vendors: MCA-AVLQSGFR-K(DNP)-K-NH2 (R & D Systems, Minneapolis, MN, USA), Z-RLRGG-AMC (Bachem Holding AG, Basel-Landschaft, Switzerland), Boc-VPR-AMC (Sigma-Aldrich, St. Louis, MO, USA), Boc-QAR-AMC (Peptides International, Inc., Louisville, KY, USA), and Z-FR-AMC (R & D Systems, Minneapolis, MN, USA).

### 4.4. Enzymatic Inhibition Assay

The protease enzymatic activities of SARS-CoV-2 Mpro and PLpro were measured using a FRET-based peptide substrate: MCA-AVLQSGFR-K(DNP)-K-NH2 and fluorogenic substrate: Z-RLRGG-AMC, respectively. The protease enzymatic activities of human thrombin, human TMPRSS2, and human cathepsin L were performed using the fluorogenic substrates Boc-VPR-AMC, Boc-QAR-AMC, and Z-FR-AMC, respectively. 

The SARS-CoV-2 Mpro (50 nM final concentration) enzymatic reaction was carried out in reaction buffer containing 50 mM Tris-HCl pH 7.5, 150 mM NaCl, 1 mM EDTA, and 0.01% Tween 20 using 10 µM of MCA-AVLQSGFR-K(DNP)-K-NH2 FRET-based peptide as a substrate. SARS-CoV-2 PLpro (24.46 nM final concentration) enzymatic assays were performed in reaction buffer containing 50 mM HEPES pH 6.5, 150 mM NaCl, and 0.01% Tween 20 using 50 µM of Z-RLRGG-AMC fluorogenic substrate. The positive control for these assays was 10 µM Ebselen. Human thrombin (678.3 nM final concentration) enzymatic assays were carried out in 50 mM Tris, 10 mM CaCl_2_, 150 mM NaCl, 0.05% Brij-35, pH 7.5 using 100 µM Boc-VPR-AMC as the substrate and 100 µM of dabigatran as a positive control. Human TMPRSS2 (at a 30 nM final concentration) was assayed in 50 mM Tris pH 8, 150 mM NaCl, and 0.01% Tween 20 buffer using 10 µM Boc-QAR-AMC substrate and 10 µM Nafamostat as a positive control. A human cathepsin L (at 1 nM) activity assay was performed in 50 mM MES, 5 mM DTT, 1 mM EDTA, and 0.005% (*w*/*v*) Brij-35, pH 6.0 using 35 µM Z-FR-AMC as the substrate and 10 µM of E64 as a positive control. The negative controls for all assays were 0.2% DMSO. All experiments were assayed in a black 384-well microplate (BD Falcon, Dhaka, Bangladesh) in tahe total volume of 50 µM at 37 °C. For the Mpro FRET-based peptide substrate, the fluorescence signals were monitored at wavelengths of 320 and 400 nm for excitation and emission, respectively. The fluorescence signals of the PLpro fluorogenic substrate were monitored at wavelengths of 360 and 460 nm for excitation and emission, respectively. All fluorescence signals were detected using a Synergy HTX Multi-Mode Microplate Reader (BioTek) and data were visualized using Gen5 Software (BioTek, Hong Kong). The dose–response curves of each compound against all selected proteases were performed on 10 concentrations in triplicate ranging from 50 mM to 100 nM and the IC_50_ values of each compound were calculated accordingly using the SciPy and Matplotlib Python packages.

### 4.5. Cells Culture and Immunofluorescence Assay

Huh-7.5.1 cells were derived from the Huh-7.5 GFP-HCV replicon cell line I/5A-GFP-6 (PMID: 15220413), kindly provided by Charles Rice (Rockefeller University, New York, NY, USA) (PMID: 15939869). For the infectivity assay, Huh7.5.1 cells (2000 cells/well, 384 well plate format) were used as host cells, infected with SARS-CoV-2 in a ‘multiplicity-of-infection’ value (MOI) of 1.0. Certain concentrations of compounds were spotted in the plates, followed by the Huh7.5.1 cells. Cells were incubated overnight at 37 °C with 5% of CO_2_. The cells were then infected by the virus and were incubated for 48 h at 37 °C with 5% CO_2_. The plates were then fixed with 4% PFA evaluated immunofluorescence signal for viral detection using rabbit anti-nucleocapsid (GeneTex, Irvine, CA, USA, cat# GTX135357) and anti-rabbit Alexa488 as a secondary antibody. Cells were counterstained with DAPI. The image signals were analyzed using MetaXpress software to quantify individual cells and infected cells.

## 5. Conclusions

A single chemical substance may specifically inhibit several enzymes with an active-site cysteine essential for COVID-19, including two viral proteases, Mpro and PLpro, and at least one human protease involved in the viral infectivity cycle. This compound may also be relatively specific to the chosen viral and host targets. We identified the derivative of thiuram disulfide or dithiobis-(thioformate) with such favorable properties with the potential to slow down the viral infectivity cycle and modulate inflammatory responses.

## Figures and Tables

**Figure 1 pharmaceuticals-15-00744-f001:**
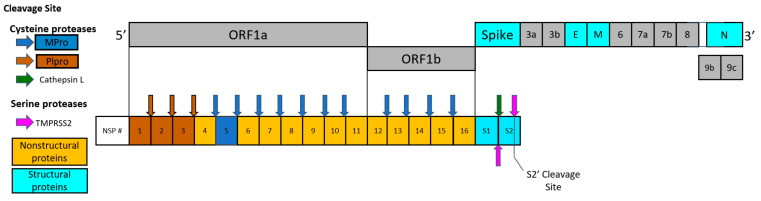
Schematic drawing of polypeptides expressed by SARS-CoV-2, including nonstructural proteins (NSPs), structural proteins, and accessory proteins. Each represented viral protein is not drawn to actual scale. Brown and blue arrows represent the cleavage sites of SARS-CoV-2 Mpro and PLpro, respectively. Cathepsin L cleavage between S1/S2 subunits is represented by the green arrow. The cleavage sites of TMPRSS2 located between S1/S2 subunits and S’ cleavage site in S2 subunit are indicated by magenta arrows.

**Figure 2 pharmaceuticals-15-00744-f002:**
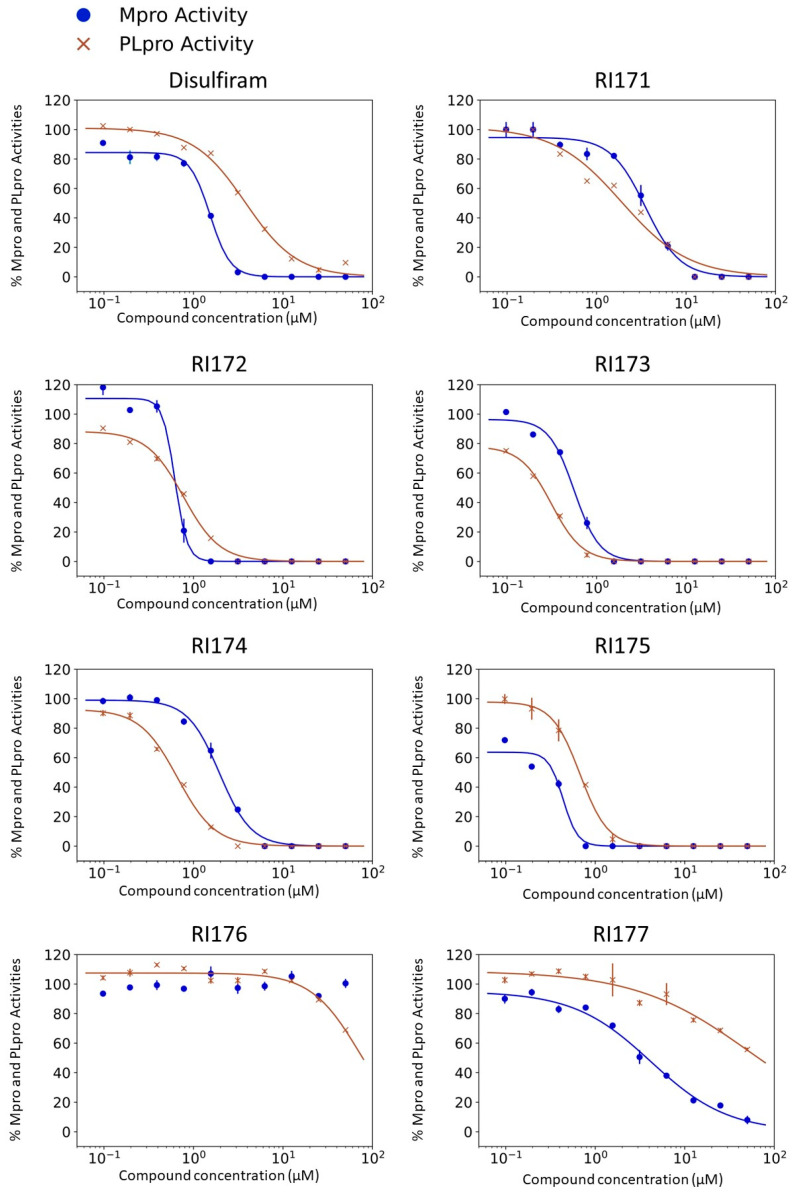
Selected thiuram disulfide or dithiobis-(thioformate) analogues show potent inhibition against SARS-CoV-2 Mpro and PLpro. Dose–response curves of disulfiram, RI171, RI172, RI173, RI174, RI175, and RI177 against SARS-CoV-2 Mpro (blue) and PLpro (brown). The disulfiram dose–response curve performed under the same conditions is shown for comparison. The proteolytic activities of both enzymes were determined by the enzymatic fluorescence assay and shown as percent activities relative to the negative control (DMSO). Error bars represent the standard errors of three independent experiments.

**Figure 3 pharmaceuticals-15-00744-f003:**
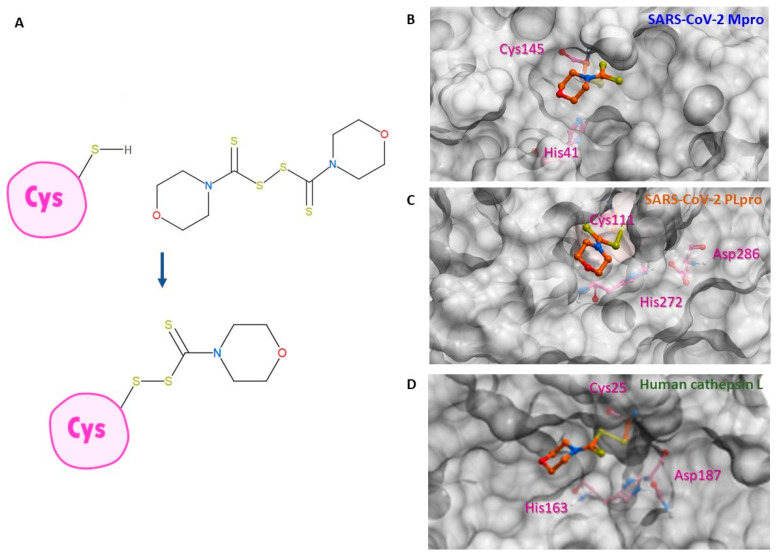
Binding conformation of thiuram disulfide or dithiobis-(thioformate). (**A**) The mechanism of RI173 when it covalently reacts to the active cysteine residue of SARS-CoV-2 Mpro and PLpro. (**B**) Predicted 3D conformation of RI173 shown as the orange stick in SARS-CoV-2 Mpro surrounded by catalytic dyads active residues Cys145 and His41. (**C**) Predicted 3D conformation of RI173 shown as the orange stick in SARS-CoV-2 PLpro surrounded by the catalytic triads active residues Cys111, His272, and Asp286. (**D**) Predicted 3D conformation of RI173 shown as the orange stick in human cathepsin L surrounded by catalytic triads active residues Cys25, His163, and Asp187.

**Figure 4 pharmaceuticals-15-00744-f004:**
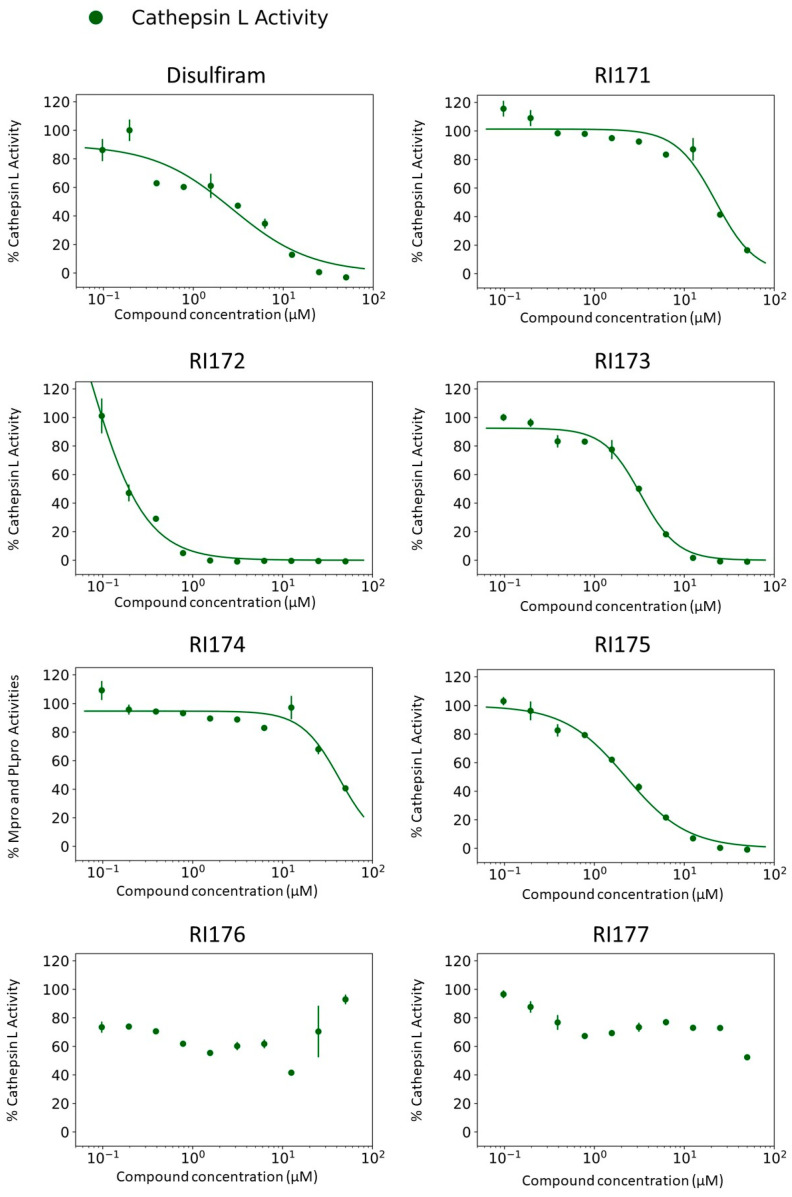
Thiuram disulfide or dithiobis-(thioformate) analogues inhibit human cathepsin L. Dose–response curves of newly identified dual viral protease inhibitors disulfiram, RI171, RI172, RI173, RI174, RI175, RI176, and RI177 against human protease cathepsin L. Error bars represent the standard errors of independent three experiments. The proteolytic activities of both enzymes were determined by the enzymatic fluorescence assay and shown as percent activities relative to the negative control (DMSO).

**Figure 5 pharmaceuticals-15-00744-f005:**
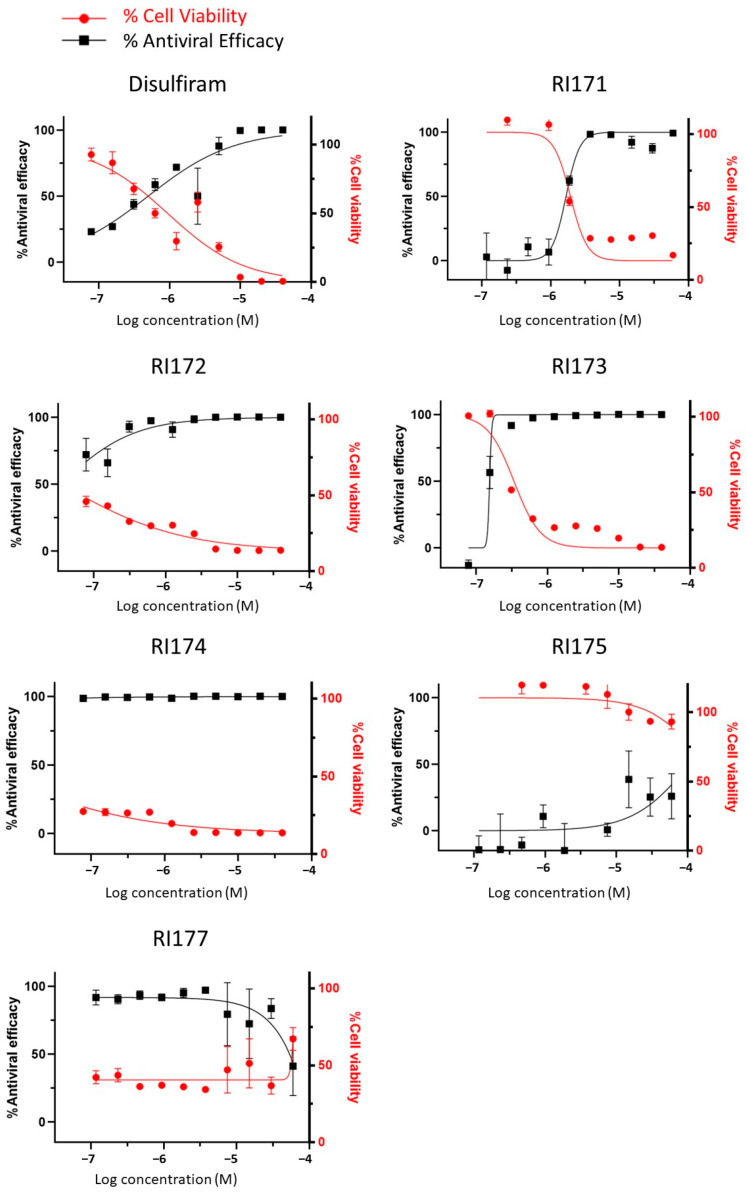
The antiviral efficacy and toxicity level of thiuram disulfide or dithiobis-(thioformate) analogs in the SARS-CoV-2 Huh7.5.1 cell-based infectivity assay. The antiviral activity (black solid square) against SARS-CoV-2 and cell viability (solid red circle) of disulfiram, RI171, RI172, RI173, RI174, RI175, and RI177 at 60 to 0.080 µM obtained from the immunofluorescence signal and normalized compared to the untreated control (0% inhibition) and non-infected control (100% inhibition and 100% cells viability). The half maximal effective concentration (EC_50_) and cytotoxic concentration (CC_50_) of each compound were evaluated from the antiviral efficacy and cell viability normalized signals. Fitting was performed using GraphPad Prism software.

**Table 1 pharmaceuticals-15-00744-t001:** Structures of the tested compounds with their known targets, the oral lethal dose 50 (LD_50_) in mice, and enzymatic assays against SARS-CoV-2 Mpro, PLpro, and human cathepsin L investigated in this study. Numbers in blue, orange, and green show the half-maximal inhibitory concentration (IC_50_) of the tested compounds against SARS-CoV-2 Mpro, PLpro, and human cathepsin L, respectively.

	Name	Structure	Previously Characterized Targets	SARS-CoV-2 Proteases Targets		Human Protease Target	LD_50_ Mouse oral in mg/kg	SARS-CoV-2 Infectivity Cells-Based Assay
				** Mpro IC_50_ (µM) **	** PLpro IC_50_ (µM) **	** Cathepsin L IC_50_ (µM) **		**EC_50_ (nM)** **CC_50_ (nM)** **CC_50_/EC_50_ Ratio**
1	Disulfiram	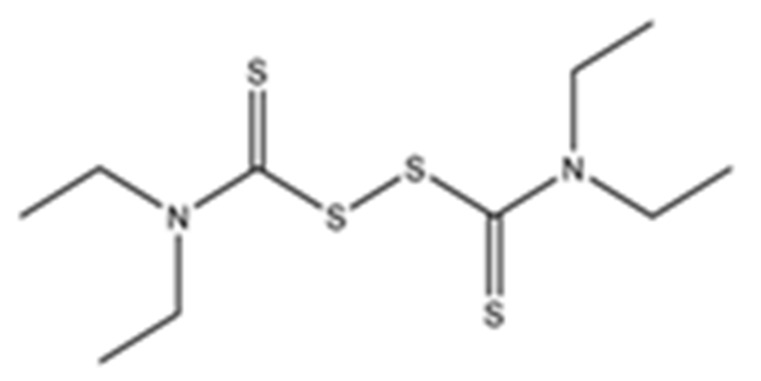	Aldehyde dehydrogenase 2 [16] SARS-CoV-2 Mpro and PLpro [18] Gasdermin D [21] Dopamine beta hydroxylase [26] Pyruvate Dehydrogenase Kinase 1 [27] Ubiquitin E3 Ligase Breast Cancer-Associated Protein 2 [28]	1.5 ± 0.1	3.8 ± 0.1	2.2 ± 1.4	1980 [29]	0.5 ± 0.2 0.9 ± 0.3 1.8
2	RI173 (JX 06) di-**Morpholine**-thiuram disulfide	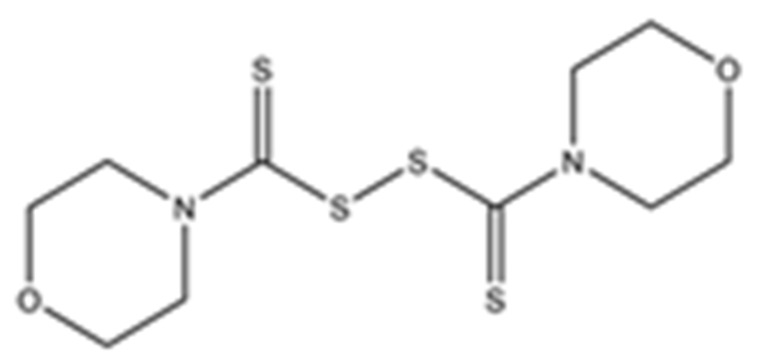	Gasdermin D [21] Pyruvate Dehydrogenase Kinase 1 [27] Human Monoacylglycerol Lipase [30]	0.6 ± 0.1	0.2 ± 0.1	1.7 ± 0.6	3250 [31]	0.1 ± 0.00 0.3 ± 0.1 3.0
3	RI175 O,O-di-**Ethyl** dithiobis-thioformate	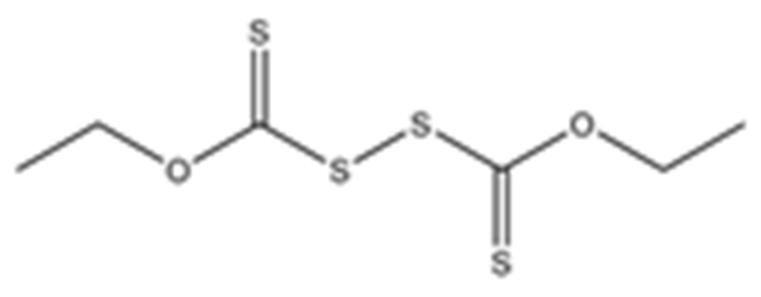		0.3 ± 0.1	0.6 ± 0.1	1.9 ± 0.6	1200 [32]	>60 >60 N/A
4	RI172 di-**Pyrrolidine**-thiuram disulfide	** 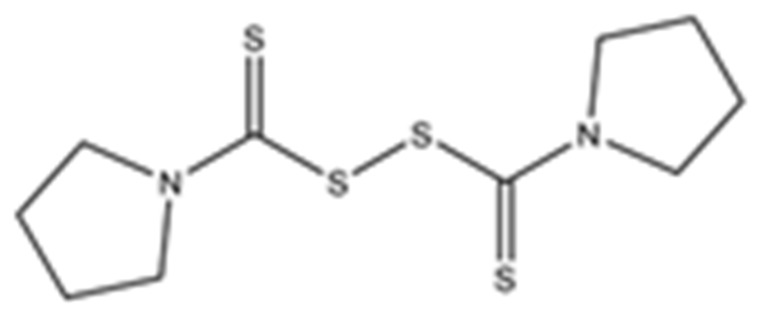 **	Ubiquitin E3 Ligase Breast Cancer-Associated Protein 2 [28] Human Monoacylglycerol Lipase [30]	0.6 ± 0.1	0.7 ± 0.0	0.2 ± 0.0	LD_50_ not available	<0.08 <0.08 Not available
5	RI174 (Thiram) tetra-**Methyl**-thiuram disulfide	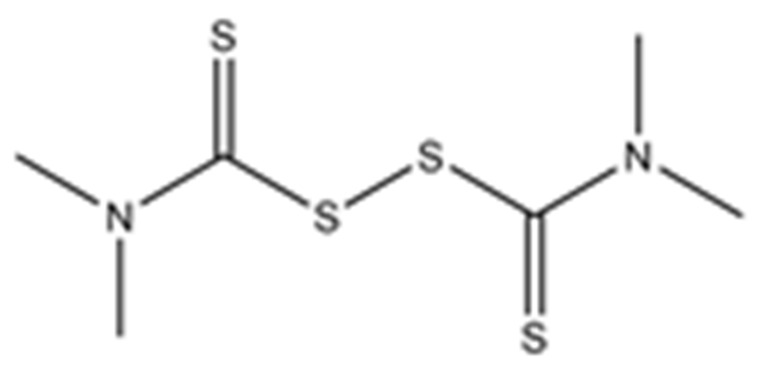	Pyruvate Dehydrogenase Kinase 1 [27]	2.0 ± 0.3	0.6 ± 0.0	16.9 ± 6.9	1350 [33]	<0.08 <0.08 Not available
6	RI171 di-**4-Methylpiperazine**-thiuram disulfide	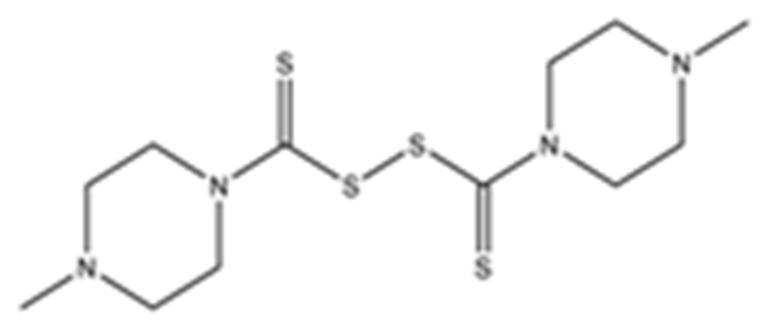	Ubiquitin E3 Ligase Breast Cancer-Associated Protein 2 [28] Human Monoacylglycerol Lipase [30]	3.2 ± 0.1	1.9 ± 0.1	4.5 ± 0.7	100 [26]	1.9 ± 0.2 1.7 ± 0.4 0.9
7	RI177 tetra-**IsoPropyl**-thiuram disulfide	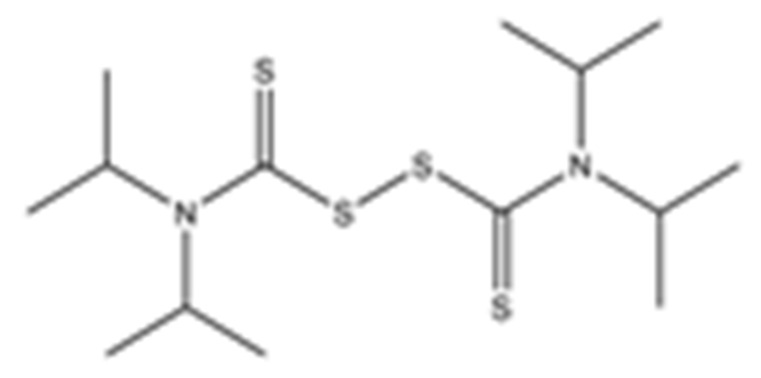		3.8 ± 0.6	>50	>50	LD_50_ not available	>60 >60 Not available
8	RI176 di-**Methyl**-di-**Phenyl**-thiuram disulfide	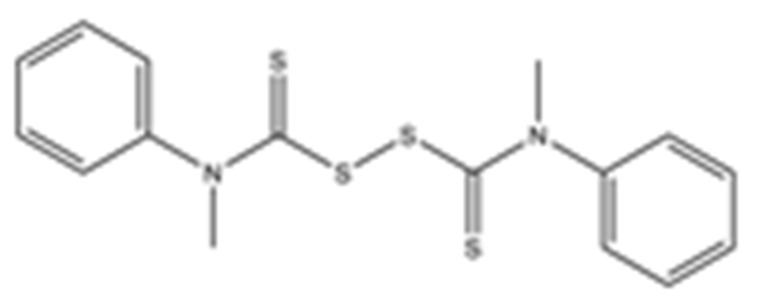		>50	>50	>50	LD_50_ not available	EC_50_ not tested CC_50_ not tested Not available

## Data Availability

Data is contained within the article and Appendix A.

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
