# Peer review of "Discovery of Triple Inhibitors of Both SARS-CoV-2 Proteases and Human Cathepsin L"

_pharmaceuticals, 2022, doi:10.3390/ph15060744_

Round 1

Reviewer 1 Report

      This is an interesting work trying to expand with related inhibitors the previous observation that the drug disulfiram could be used to treat viral infection with SARS-CoV-2 through covalent inhibitory action on three cysteine proteases required for the entry and replication of the virus in mammalian cells. The work is attractive and well done regarding the analysis, derivation and assay of a set of selected inhibitors on the proteases Mpro, PLpro and cathepsin L, the former two belonging to the virus and the third one to the host mammalian cells.  However, there are a series of points in the work either not sufficiently explained or checked which should be improved to give enough consistency and trust to the proposal of such selected inhibitors as lead compounds with such pharmacological goal.

1.-What is the degree of specificity of these compounds?  Disulfiram, a currently approved drug as an aversive treatment of chronic alcoholism, through its covalent inhibitory action on the highly reactive active site cysteine of an aldehyde dehydrogenase, is in Clinical trials for the Covid-19 treatment because it seems to produce an equivalent action on the reactive cysteines of the above mentioned proteases involved in SARS-Co-V2 infection (and on other viral cysteine proteases).  The authors expect that the newly derived inhibitors, carrying the same thiuram disulfide or an equivalent dithiobis-(thioformate) reactive group/warhead display better inhibitory capability towards such triplet of cysteine proteases, including human cathepsin L, than disulfiram, and certainly this happens with some, particularly with compound RI173 which, at the same time shows the lower cell toxicity of all of them. However, the differences are not impressive, with substantial toxicity of all, and there is no guarantee that such compounds will be specific enough and not toxic when assayed in vivo.  In fact, even at the in vitro o in cellulo levels here assayed, nothing has been examined or specifically mentioned regarding the possible action of such compounds on other classes of enzymes with reactive cysteines, besides generic warnings. This includes the known targets for disulfiram, aldehyde dehydrogenases, important at the metabolic and detoxification levels, among others. Authors should at least experimentally test the effect of such compounds on these aldehyde dehydrogenases, optimally on several variants of them, or at least with the one more related with disulfiram, ALDH2.

    By the way, in two locations of the manuscript the term "alcohol dehydrogenase" has been used incorrectly instead of "aldehyde dehydrogenase":  at lines 88 and 320.

2.-Why the SARS-CoV-2 Huh 7.5.1 infection model has been used in this work?  African green monkey kidney Vero cells are commonly used to propagate and study SARS-CoV-2 (Cagno V., 2020; Chu H. et al., 2020, Lancet Microbe). Why the Huh-7 cells have been used here?  This should be explained in the manuscript, as well as provide more information about them, including the supplier.

3.-What is the purpose of Fig. 3?  The reasons of including Fig. 3 are not obvious from section 2.2 and 4.1 in which such figure is commented.  it would be worth to improve them about.

4a.-The mention, at line 30, that RI173 is  ", a protease inhibitor,", is unnecessary because this is evident from the previous lines.

4b.-Several of the publications collected in the References section lack authors' names or are uncompleted, like numbers 2, 35, 37, 45. 

4c.-Besides, ref. 18, that was provisionally archived at the bioRxiv journal, have been later published at "ACS Pharmacol Transl Sci. 2020 Oct 9;3(6):1265-1277.doi: 10.1021/acsptsci.0c00130.".

Reviewer 2 Report

COVID pandemic has set the priority  in identifying effective antiviral drugs to tackle SARS-CoV-2 infection. Main Protease (Mpro) and Papain-like Protease (PLpro) encoded by the viral genome are already known as important targets. In order to limit overall SARS-CoV-2 viral replication, it is becoming clear that the inhibition of both proteases has obvious benefits than inhibit the individual protease. The goal of the authors was to target multiple proteases that are necessary for the viral replication process with a single small molecule inhibitor. They explored the activity of several analogs of disulfiram, currently in Phase 2 Clinical trials, to determine more effective inhibitors that take advantage of disulfiram’s mechanism of binding to an active-site cysteine. Although they identified only one compound that shows a small improvement from the approved drug, these results provide the proof of concept of targeting multiple functional proteins involving SARS-CoV-2 viral replication using thiuram disulfide or dithiobis-(thioformate) compounds for the improvement of COVID-19 treatment.

Specific comments :

Line 66-74: The sentences are redundant, I would suggest to reformulate in order to provide the information in a more readable way.

Line 123-130: The text is redundant, as authors have already described this in the introduction. I would make it more concise and start explaining the results.

Line 168: Could authors please explain better the implications of the results obtained in presence of DTT? Do they think that the susceptibility of these new compounds to nonspecific competing cysteine compounds could affect their activity in case of systemic administration?

Line 182: SARS-CoV-2 can be activated by cathepsin L in the endosome while TMPRSS2 is localized on the cell surface . Therefore results showing the inhibitory activity of the compounds on CtsL are not demonstrating the capability of inhibiting cell entry. I would suggest to modify the title of the paragraph (“Cell entry proteins inhibition”) considering that the tested compounds are not active against TMPRSS2.

Figure S2: Please adjust the graph scale, it is really difficult to interpret The results.

Reviewer 3 Report

The authors presented a study on the SARS CoV-2 Main Protease inhibitors. All the theoretical hypotheses raised by the authors are experimentally supported. 

I have one minor remark regarding the layout of Table 1, which is difficult to read in the current form.

Author Response

This manuscript is a resubmission of an earlier submission. The following is a list of the peer review reports and author responses from that submission.

Round 1

Reviewer 1 Report

Scientific inspiration – based on disulfiram- of the disclosed inhibitor warhead in this manuscript is proper

Authors may need to disclose 1) complete synthesis, 2) 1H NMR, 3) 13C NMR, 4) HRMS data, 5) HPLC purity traces that show more than 95% purity, 6) Melting points if the compounds are solid, 7) physical appearance of the compounds, preferably for all the intermediates and final compounds or at least for the final compounds, if they are not reported elsewhere, given that the context of the manuscript describes a SAR. If the inhibitor data 1-7 are reported elsewhere, it should be clearly noted in the manuscript directing the readership to collect that information 1-7.

Resolution of the figures needed to be improved to visualize the figure descriptions, numbers, and legends clearly

Curve fitting data needed to be disclosed, showing R2 values preferably are more than or equal to 0.9 for all the curves and SD. It’s appropriate to provide raw assay data in the supplementary section for the readership and the transparency

Authors may need to be descriptive about particularly very high cytotoxicity (CC50) and possible off-target effects of the reported compounds to that of disulfiram and already reported safe and efficacious other SARS-CoV-2 main and Papain-like protease inhibitors (e.g., Paxlovid, GC376, etc.)

Reviewer 2 Report

This manuscript reports the screening and evaluation of a series of thiuram disulfide derivatives as multi-targeting inhibitors against proteases/enzymes for potential treatment of COVID-19. This work focused on improving the efficiency of and reducing the toxicity of disulfiram, which also is a thiuram disulfide derivative and clinically used to treat chronic alcoholism, and currently in phase II clinical trials against SARS-CoV-2. This work were performed competently, and demonstrated that three compounds, namely RI175, RI172, and RI172 are the most potent triple inhibitor in vitro against the SARS-CoV-2 proteases, Mpro and PLpro, and the human cathepsin L. Cell-based assays indicated that RI173 has greater antiviral effect, and less toxicity to cells than disulfiram, and deserves further development for treatment of the COVID-19 virus or related variants infection. However, I do not think that this manuscript in its current form is suited for publication in Pharmceutics.

My most concern is about the presentation style of this work. I noticed that all the compounds studied in this work are commercially available, and have been studied for screening effective inhibitors for other proteases and enzymes containing cysteine catalytic sites, such as gasdermin D, pyruvate dehydrogenase kinase 1 and monoacylglycerol lipase. However, the authors did not mention this background information at all in the introduction section. The reviewer has not realized that all the compounds were commercially available until reading the summary for the previously characterized targets of the studied compounds in Table 1. Moreover, the authors stated “disulfiram and RI173, our best compound, is a known inhibitor of pyruvate dehydrogenase kinase 1”, again making readers misunderstand that all compounds studied in this work were developed in authors’ lab. In my opinion, the authors should make it clear in Introduction that all compounds have been reported previously as effective inhibitors against other proteases or enzymes containing cysteinyl function group, these inspired them to select these compounds for the present study. Additionally, the reviewer noticed that three compounds, RI172, RI173 and RI174, were named as JX07, JX06 and JX03, respectively, in Cancer Res. 2015 Nov 15;75(22):4923–36 which is cited as ref. 31 in this manuscript, and wonder what is the fundament for naming the selected compounds in this work.

Other specific comments:

  1. The content of Conclusions is not conclusions, but the description of methods for measurement of enzymatic activities. Please correct;
  2. Line 167, “Fig S” should read as “Fig. S1”;
  3. In Table 1, “(GSDMD) (18)” should read as “(GSDMD) (19)”;
  4. In the caption of Fig. S2, “human TMPRSS2” were mistakenly written as “Cathepsin L.”.

Reviewer 3 Report

In the work by R. Abagyan et al, although compounds show cell toxicity it is worthy to be published and can be interesting for the scientific community. It is well written but some changes should be made in the manuscript as detailed below.

-A figure at the beginning of the manuscript depicting chemical structure of disulfiram and related inhibitors  can be added for a better understanding by the reader (probably in page 2 or 3). It could also include a short scheme of the mode of action by reacting with thiol of catalytic cysteine.

-The chemical structures and graphs of figures and tables are of very low quality and should be changed by better ones.

-Docking studies have been performed and a figure is indicated but there is no explanation of it in the text. A short summary of that part should be included.

Reviewer 4 Report

Abagyan and co-workers present disulfiram analogs as triple inhibitors of SARS CoV-2 two proteases (Papain Like and Main Protease) and human protease, cathepsin L. The authors have screened derivatives of disulfiram, a previously reported SARS-CoV-2 Mpro, and PLpro inhibitor, against cysteine proteases.  Using in vitro enzymatic assays, they have demonstrated that disulfiram analogs have either similar or moderately improved inhibitory activities against Mpro and PLpro compared with disulfiram. Although disulfiram analogs showed promising activity in enzymatic assays, these compounds were found to be either too toxic to host cells or failed to exhibit antiviral activity. A major weakness of the study is a narrow selectivity index. Since the goal of this study was to identify analogs with improved efficacy and reduced toxicity, it is critical to address the toxicity issue with additional compounds. Generally, the experiments were well designed and performed. The inhibitors presented are interesting and merit publication in Pharmaceutics after addressing the below comments.

  1. In table 1, the order of compounds looks confusing. The authors should arrange the compounds according to either increasing steric bulk at the nitrogen of dithioamide or compound numbers.
  2. Can the authors comment on why these compounds did not inhibit TMPRSS2 protease? 

  3. Page 4- "The replacement of ethylamine groups in disulfiram with morpholine, pyrrolidine, methylamine, and 4-methylpiperazine in RI173, RI172, Thiram, and RI171, respectively, moderately enhance the potency of inhibitors against Mpro and PLpro, while substitution by isopropylamine in RI177 showed the preservation of inhibition of Mpro but completely lost the inhibitory activity (IC50 > 50 μM) against PLpro." In this sentence, ‘ethylamine’ should be replaced with ‘diethylamine’. Similarly, ‘methylamine’ and ‘isopropylamine’ should be replaced with ‘dimethylamine’ and ‘diisopropylamine’, respectively.

  4. Page 9- The authors stated that “Previously, it was shown that all thiuram disulfide or dithiobis-(thioformate) compounds have high potency against Mpro and PLpro of SARS-CoV-2 and human cathepsin L.” This statement seems to be incorrect because according to Table 1, RI176 and RI177 were not effective against Mpro and PLpro of SARS-CoV-2 and human cathepsin L.

  5. Page 11- “However, compounds RI172, Thiram, and RI177 were too toxic for the host cells in the selected range suggesting the CC50 and EC50 values should be lower than 80 nM.” However, according to Table 1, IC50 and CC50 of RI177 are >60 uM. 

  6.  

    The figures in the manuscript look blurred and hard to read. The authors should increase the resolution of the figures. Also, the font size of the figure label is too small.  

  7.  

    4.4. Enzymatic inhibition assay- the authors have repeated the following information from the compounds and reagents section. “Disulfiram was purchased from Fisher Scientific (Hampton, NH). RI171 and RI172 were purchased from MolPort (Latvia) while RI173 was purchased from Ambeed (Arlington Heights, IL). Thiram.” This needs to be deleted. 

  8.  

    Conclusions- The authors have essentially added the material and methods part under the conclusions. This section needs some attention, and the authors should discuss the major findings.  

Reviewer 5 Report

The manuscript entitled “Discovery of Triple Inhibitors of Both SARS-CoV-2 Proteases 2 and Human Cathepsin L” reports on the evaluation of thiuram disulfide derivatives against the two SARS-CoV-2 proteases (Mpro and PLpro) as well as against the human cathepsin L. In particular, starting from disulfiram, which is a drug used for the treatment of chronic alcoholism and actually in phase 2 clinical trials as anti-SARS-CoV-2 agents, in the present work authors focused on the identification of disulfiram derivatives with anti-SARS-CoV-2 activity by interfering with both SARS-CoV-2 proteases and also the human cathepsin L possibly endowed with more potent efficacy than disulfiram in order to reduce its side-effects.

Although the significance of the work could be high and important, the results obtained by authors in the present work are not considerable enough to deserve publication in Pharmaceutics. Indeed, starting from the already reported SARS-CoV-2 proteases inhibitory activity of disulfiram, authors collected only six structurally related disulfiram derivatives commercially available. Although some compounds, such as RI173, showed a slight better in vitro efficacy than disulfiram in inhibiting SARS-CoV-2 proteases and cathepsin L, no compound showed a better behaviour in terms of cytotoxicity. Indeed, RI173 has a SI of 1.5 (and not 2.3 as reported by authors), even worse than disulfiran (SI = 1.8). I strongly suggest authors to explore more thiuram disulfide derivatives in order to gain more insights to delineate a clear SAR but above all identify less toxic derivatives endowed with improved SI values.

Minor comments:

  • An overview on the current anti-COVID drugs and most advanced candidates is missing.
  • More recent reviews collecting Mpro and PLpro inhibitors in preclinical and clinical study can be added in the introduction.
  • In table 1 several errors are present: in the first line, for EC50 and CC50 nM concentration is indicated while, based on the values reported in the text, the numbers are expressed in uM; the “ratio” used by authors is usually indicated as Selectivity Index and calculated by the ratio between CC50 and EC50. Based on the number reported in table 1, the SI for almost all the compounds are wrong. Please check and correct.
  • Figure 3 caption: in the Figure is reported compound RI173 and not RI172 as authors indicated.
  • Conclusions must be re-written. The conclusion paragraph seems more a paragraph related to material and methods and not the conclusions of the work done.